# Exogenous Melatonin Reduces Lignification and Retains Quality of Green Asparagus (*Asparagus officinalis* L.)

**DOI:** 10.3390/foods10092111

**Published:** 2021-09-07

**Authors:** Athip Boonsiriwit, Myungho Lee, Minhwi Kim, Pontree Itkor, Youn Suk Lee

**Affiliations:** 1Department of Packaging, Yonsei University, Wonju 220-710, Korea; athip8266@gmail.com (A.B.); yb8049@naver.com (M.L.); minhwi10@gmail.com (M.K.); pontree.itkor@gmail.com (P.I.); 2Rattanakosin International College of Creative Entrepreneurship (RICE), Rajamangala University of Technology Rattanakosin, Nakhon Pathom 73170, Thailand

**Keywords:** asparagus, melatonin, shelf life, lignification, post-harvest treatment

## Abstract

Asparagus (*Asparagus officinalis* L.) is highly perishable because of its high respiration rate, which continues after harvesting and leads to weight loss, increased hardness, color change, and limited shelf life. Melatonin is an indoleamine that plays an important role in abiotic stress. This study was designed to investigate the effects of melatonin on the quality attributes of green asparagus during cold storage. Green asparagus was soaked in a melatonin solution (50, 100, and 200 μM) for 30 min and then stored at 4 °C under 90% relative humidity for 25 days. The results indicated that melatonin treatment delayed the post-harvest senescence of asparagus and maintained high chlorophyll and vitamin C levels. Melatonin treatment hindered phenylalanine ammonia-lyase and peroxidase activities and reduced lignin content, thereby delaying the increase in firmness. Moreover, melatonin treatment enhanced catalase and superoxide dismutase activities, leading to reduced hydrogen peroxide content. These results indicate that melatonin treatment can be used to maintain the post-harvest quality and prolong the shelf life of green asparagus.

## 1. Introduction

The unique taste and texture of asparagus (*Asparagus officinalis* L.) have made it a popular vegetable, recognized as a rich source of nutrients and bioactive compounds with therapeutic properties, such as weight control, anti-oxidation, anticancer, radioprotection, and blood pressure control [1,2]. However, fresh asparagus has a short shelf life and a high metabolic rate and, consequently, a high post-harvest respiratory activity; this leads to weight loss, increased hardness [3,4], rapid nutrient and bioactive compound loss [4], and chlorophyll degradation [5] during storage.

The freshness of produce is one of the most important attributes that consumers consider when making purchase-related decisions. Texture is an important freshness indicator of asparagus spears and is related to their freshness. After harvesting, the firmness of asparagus increases being a general behavior of asparagus, followed by wrinkling, toughening, and hardening of the spears due to lignification [6]. Lignification is attributed to secondary cell wall formation following the upregulation of cellulose, hemicellulose, and lignin biosynthesis-related activities of enzymes such as phenylalanine ammonia-lyase (PAL) and peroxidase (POD) [6,7]. Various approaches to improving asparagus quality and extending its shelf life have been investigated, including cultivar development [8,9,10,11], agronomic management [10], post-harvest treatment [1,12,13,14], and improved packaging technology [3,15].

Melatonin (*N*-acetyl-5-methoxytryptamine), a natural low molecular weight indoleamine, was first reported in bovine pineal glands in 1958. The various roles of melatonin in numerous plants have been reported previously, including scavenging of free radicals and induction of biotic and abiotic stress resistance [16]. Many studies have recently reported that treatment of fruits and vegetables with melatonin can improve quality and shelf life. For example, melatonin treatment improved tomato quality [17], reduced decay and maintained quality in strawberries [18], delayed senescence and reduced chilling injury in peaches [16], decreased physiological deterioration of cassava [19], retarded lignification of bamboo shoots [7], delayed leaf senescence and chlorophyll degradation in Chinese flowering cabbage [20], increased the chilling tolerance of green bell pepper [21], and delayed chlorophyll degradation in broccoli, thereby extending its shelf life [22].

Asparagus is a high-value vegetable. Prolonging its freshness and shelf life by post-harvest treatment may enhance its commercial potential. To the best of our knowledge, there are no available data on the effects of melatonin on post-harvest quality aspects of green asparagus spears. Therefore, the aim of the present study was to investigate the effects of melatonin on the quality attributes of green asparagus during cold storage in terms of texture; weight loss; lignin, chlorophyll, and vitamin C contents; and PAL, POD, catalase (CAT), and superoxide dismutase (SOD) activities.

## 2. Materials and Methods

### 2.1. Raw Material

Fresh green asparagus spears (*A. officinalis* L.) were purchased from a local farm in Wonju (Gangwon-Do, Korea) and transported to the laboratory within 2 h, before chilling at 4 °C for 4 h. Subsequently, they were sorted according to the following criteria: straight, undamaged, 1.2–1.5 cm diameter, ~20 cm in length with closed bracts, and no visible signs of injury. The asparagus spears were randomly separated into sample sets of approximately 250 g each.

### 2.2. Melatonin Treatment

Melatonin 98.0% was purchased from Tokyo Chemical Industry Co., Ltd. (Tokyu, Japan). The melatonin treatment was performed at 25 °C as described by Tan et al. [20]. Asparagus spears (250 g) were immersed in 50, 100, and 200 μM melatonin solution (equivalent to 11.61, 23.22, and 46.44 mg of melatonin/L of distilled water, respectively) for 30 min, followed by air drying at 25 °C for 45 min; control samples were treated with distilled water. Subsequently, the samples were transferred to a perforated plastic basket and placed in an upright position for storage at 4 °C under 90% relative humidity. Sampling and analyses took place on day 0, 5, 10, 15, 20, and 25 of storage. The 50, 100, and 200 μM melatonin treatment groups are coded as: 0.5 MLT, 1.0 MLT, and 2.0 MLT, respectively.

### 2.3. Quality Determination

#### 2.3.1. Weight Loss

The weight of asparagus was measured after treatment on day 0 and on the aforementioned sampling days, up to day 25. Weight loss was expressed as the percentage loss of initial weight using the following formula:Weight loss (%) = (Initial weight − final weight/Initial weight) × 100(1)

#### 2.3.2. Respiration and Ethylene Production Rate

The respiration rate (Ri) of asparagus was measured using a closed-glass jar system. Briefly, asparagus (approximately 100 g) was placed in a 1 L glass jar and closed with a lid attached to the septum. The packed jars were incubated in a temperature-controlled chamber at 4 °C for 1 h. After incubation, headspace gas was measured to determine the CO_2_ level using a gas analyzer (PBI Dansensor-CheckMate II; Dansensor A/S, Ringsted, Denmark). Ri of asparagus was calculated using the following linear equation:Ri = (Y_t_ − Y_t0_) × V/(t × M × 100)(2)
where Y_t_ is the CO_2_ gas concentration at time t, M is the mass of the asparagus, and V is the volume (mL) of the headspace, according to Castelló et al. [23]. The results are expressed as mg CO_2_/kg/h.

To measure the ethylene production rate of green asparagus, the packed jars were incubated at 4 °C for 4 h. Subsequently, 1 mL of the headspace gas was collected from the jar and injected into a gas chromatograph (GC; 6890 gas chromatograph, Agilent Technologies, Inc., California, USA) equipped with an internal diameter and capillary column length of 0.32 mm and 30 m, respectively, and a flame ionization detector (FID). The column oven temperature was maintained at 50 °C for 3 min and then increased to 250 °C at a rate of 15 °C/min. The injector and FID temperatures were 100 and 250 °C, respectively. Helium was used as the carrier gas at a flow rate of 10 mL/min. The identified peak was compared with the standard curve and the retention time. The amount of ethylene gas in the headspace was calculated and expressed as µL C_2_H_4_/kg/h.

#### 2.3.3. Texture Analysis

The asparagus spears, 20 cm in length, were marked at the following three points: apical (3 cm from the tip), middle (middle of the spear), and basal (3 cm from the base). The texture was analyzed at the marked point of each of the three sections of each spear by applying the cutting test using a TA1 Texture Analyzer (Lloyd Instruments/Ametek, Largo, FL, USA) with a 1 mm diameter shear blade and a constant moving rate of 5 mm/s for a 5 mm cutting depth [3]. The maximum force (N) was expressed as the mean of 10 replicates.

#### 2.3.4. Objective Color Measurement

The external color attributes of asparagus were measured using a colorimeter (Konica Minolta Sensing, Inc., Osaka, Japan) equipped with an 8 mm measuring head and a C illuminant (6774 K) to obtain *L**, *a**, and *b** values. The device was calibrated using a standard white plate. Measurements were taken in the middle of the sample (*n* = 10 asparagus spears/treatment). The following equation was used to calculate the total color difference (Δ*E*):(3)$$\Delta E=\sqrt{{\left(L^{*}-L_{0}^{*}\right)}^{2}+{\left(a^{*\;}-a_{0}^{*}\right)}^{2}+{\left(b^{*}-b_{0}^{*}\right)}^{2}}$$
where *L*_0_*, *a*_0_*, and *b*_0_* are the initial color parameters measured on day 0 and *L**, *a**, and *b** are the color values of asparagus measured at different storage periods.

#### 2.3.5. Chlorophyll Content

Asparagus (50 g) was chopped, and 1.0 g of the chopped tissue was homogenized at a moderate speed for 30 s in 20 mL of 80% acetone. After homogenization, 80% acetone solution was used to wash the paste into a 50 mL volumetric flask and then made up to volume with 80% acetone. The homogenate was filtered through two layers of filter paper and centrifuged at 10,000× *g* for 15 min at 4 °C. The absorbance was read at 647 (Abs_647_) and 664.5 nm (Abs_664.5_) using a UV-vis spectrophotometer (V-650 Spectrophotometer; JASCO Corporation, Tokyo, Japan). The total chlorophyll content, expressed on a fresh weight (FW) basis, was calculated as follows [5]:Chlorophyll content (mg/Kg FW) = 5 × [17.95 × Abs_647_ + 8.08 × Abs_664.5_](4)

#### 2.3.6. Vitamin C Content

The vitamin C (ascorbic acid) content of asparagus, expressed as mg/100 g FW, was determined using the method of the Association of Official Analytical Chemists [24]. Briefly, a 5 g sample of 50 g of chopped asparagus was blended with 50 mL of metaphosphoric-acetic acid solution to extract ascorbic acid. The mixture was homogenized at 12,000× *g* (on ice, shielded from light) for 1 min and then centrifuged at 9000× *g* for 20 min at 4 °C. The supernatant was transferred to a volumetric flask and rapidly titrated with indophenol solution until a distinct rose-pink color persisted for more than 5 s.

#### 2.3.7. Lignin Content

The lignin content was determined using the method described by Fukushima et al. [25]. Briefly, fresh asparagus tissue (1.0 g) was homogenized with 5 mL of 95% ethanol (*v*/*v*) using an Ultra-Turrax homogenizer (IKA, Staufen, Germany) and centrifuged at 10,000× *g* for 10 min at 4 °C (2236R; Labogene, Daejeon, Korea). The residue was washed three times with 3 mL of ethanol:hexane (1:2, *v*/*v*) solution. Then, the sample was dissolved in 1 mL of acetyl bromide:acetic acid (1:3, *v*/*v*) solution and allowed to digest in a 50 °C water bath for 2 h with shaking every 30 min. After cooling and centrifugation (3000× *g* for 15 min), the digested solution (0.5 mL) was added to a tube containing 6.5 mL of acetic acid and 2.0 mL of 0.3 M NaOH. The contents were mixed, and hydroxylamine hydrochloride solution (1.0 mL) was added, followed by additional mixing of the contents, after which the absorbance of the samples at 280 nm was measured. The lignin content was calculated by comparing with a standard coumaric acid curve and expressed as g/Kg FW of asparagus.

#### 2.3.8. Hydrogen Peroxide (H_2_O_2_) Content

The concentration of H_2_O_2_ was measured colorimetrically, according to Jana et al. [26]. Briefly, 5 g of the sample was homogenized with 50 mL of phosphate buffer (50 mM, pH 6.5). Subsequently, the homogenate was centrifuged at 10,000× *g* for 30 min at 4 °C. Three milliliters of the supernatant was mixed with 1 mL of 0.1% titanium sulfate in 20% (*v*/*v*) H_2_SO_4_. The absorbance of the sample at 480 nm was recorded. The H_2_O_2_ content was calculated by comparing with a standard H_2_O_2_ curve and expressed as mg/100 g FW of asparagus.

#### 2.3.9. Enzyme Activity Assays

PAL activity was measured as described by Wang et al. [27]. Briefly, fresh asparagus tissue samples (2.0 g) were ground with 6 mL of 50 mM Tris-HCl buffer (pH 8.8) containing 15 mM β-mercaptoethanol, 5 mM ethylenediaminetetraacetic acid (EDTA), 5 mM ascorbic acid, and 0.15% (*w*/*v*) polyvinylpyrrolidone (PVP). The homogenate was centrifuged at 12,000× *g* for 30 min at 4 °C. The supernatant was used as a source of crude enzyme to assay PAL activity. The reaction mixture (3 mL), containing 16 mM l-phenylalanine, 50 mM Tris-HCl buffer (pH 8.9), 3.6 mM NaCl, and crude enzyme extract (0.5 mL), was incubated at 37 °C for 1 h; the reaction was stopped by adding 500 µL of 6 M HCl. PAL activity, expressed as U/mg protein, was determined by monitoring the increase in absorbance at 290 nm.

POD activity was measured according to the method described by Zhang et al. [28]. Two grams of the sample was homogenized with 10 mL of 0.05 mM sodium phosphate buffer (pH 7.0) containing PVP (0.2 g). The homogenate was then centrifuged at 12,000× *g* for 30 min at 4 °C; the supernatant was collected as the crude enzyme extract. The POD reaction mixture contained 1 mL of guaiacol, 0.25 mL of enzyme extract, and 1.75 mL of phosphate buffer (pH 7.0). POD activity, expressed as U/mg protein, was determined by monitoring the increase in absorbance at 470 nm.

CAT activity, expressed as U/mg protein, was measured using the method described by Li et al. [29] with some modifications. Briefly, 2 g of sample was homogenized with 10 mL of 0.2 M phosphate buffer (pH 7.8) containing 1% (*w*/*v*) PVP and then centrifuged at 12,000× *g* for 30 min at 4 °C; the supernatant was collected as the crude enzyme extract. The CAT reaction contained 0.2 mL of crude enzyme, 0.3 mL of 0.1 M H_2_O_2_, 1.5 mL of phosphate buffer (pH 7), and 1 mL of distilled water. CAT activity was determined by monitoring the decline in absorbance at 240 nm, reflecting the decrease in H_2_O_2_ content.

SOD extraction was performed using the method described by Li et al. [29]. Two grams of fresh asparagus tissue was homogenized with 10 mL of 50 mM phosphate buffer, containing 0.1 mM EDTA, 0.3% (*w*/*v*) TritonX-100, and 4% (*w*/*v*) PVP. The homogenate was then centrifuged at 12,000× *g* for 30 min at 4 °C; the supernatant was collected as the crude enzyme extract. SOD activity was measured according to Tang et al. [30] and expressed as U/mg protein.

### 2.4. Statistical Analysis

Statistical analysis was performed using IBM SPSS Statistics for Windows (Version 24.0.; IBM, Armonk, NY, USA). Firmness and color analyses were conducted in 10 replicates, whereas the other analyses were conducted in triplicate. Duncan’s multiple range test was used to determine significant differences at *p* < 0.05. Different superscript lowercase letters are used for comparing significant differences in the same column while different superscript uppercase letters are used for comparing significant differences in the same row (*p* < 0.05). Data are displayed as the mean ± standard deviation.

## 3. Results and Discussion

Asparagus quality changes rapidly after harvesting because of the high Ri. Typically, asparagus spears have a shelf life of 3–5 days under ambient temperature storage. Post-harvest physiological, chemical, and biological compositional changes that reduce asparagus quality include bract opening, toughening, weight loss, chlorophyll degradation, and vitamin C loss.

### 3.1. Effect of Melatonin Treatment on Weight Loss, Respiration, and Ethylene Production Rate

Water transpiration has a critical effect on several physiological reactions in asparagus. Moreover, asparagus water loss is also crucial in terms of marketing; weight loss <6% during asparagus spear storage is considered acceptable [31]. Asparagus weight loss is primarily attributed to differences in the water vapor pressure between the spear surface and the atmosphere. Herein, the weight loss of asparagus in all treatment groups increased gradually during storage (Figure 1a). All the melatonin-treated asparagus spears presented with significantly decreased weight loss compared with the control from day 15 until the end of the storage period. The weight loss of the control group after 20 days (6.18 ± 0.61%) exceeded the rejection limit (i.e., weight loss >6%), whereas 0.5 MLT and 2.0 MLT exceeded the rejection limit after 25 days. Notably, 1.0 MLT did not exceed the weight loss limit during the entire storage period. At the end of the storage period, the control group weight loss amounted to 9.06 ± 0.41% whereas that of 0.5 MLT, 1.0 MLT, and 2.0 MLT amounted to 6.83 ± 0.59, 5.66 ± 0.19, and 6.01 ± 0.49%, respectively. These results indicate that melatonin treatment delays water loss by 5 days. Treatment with 1.0 mM melatonin showed the best results because the weight loss was still within the acceptable range at the end of the storage period.

Respiration of fresh produce is related to product deterioration, nutrient loss, and short shelf life [32]. In this study, the initial Ri (Figure 1b) of all treatment groups was the same, i.e., in the range of 16.34–17.53 mg CO_2_/kg/h. During storage, Ri showed the same pattern in all groups, increasing gradually until day 20 before subsequently decreasing. 0.5 MLT displayed significantly lower Ri than the control after 15 days of storage, whereas Ri of 1.0 MLT and 2.0 MLT was significantly lower than that of the control from day 5. At the end of storage (day 25), the control group Ri was 29.42 ± 1.93 mg CO_2_/kg/h whereas that of 0.5 MLT, 1.0 MLT, and 2.0 MLT was 22.89 ± 2.20, 18.19 ± 1.30, and 20.95 ± 1.43 mg CO_2_/kg/h, respectively. The lower Ri of melatonin-treated asparagus may be attributed to attenuation of the Embden–Meyerhof–Parnas pathway, which is related to the generation of ATP and NADH during glucose metabolism, resulting in a reduced Ri [33,34]. Similar results have been reported for Chinese flowering cabbage [33], apples [35], cherries [36], and mushrooms [37].

The ethylene production rate increased in all treatment groups with increasing storage periods (Figure 1c). On day 0, there were no significant differences observed between the ethylene production rate of the control and that of the melatonin-treated asparagus (value range = 0.11–0.14 μL/kg/h). However, the ethylene production rate of all melatonin treatment groups was significantly lower than that of the control group from day 5. The ethylene production rate of 0.5 MLT, 1.0 MLT, and 2.0 MLT was 17.14, 30.58, and 29.65% lower than that of the control group, respectively, at the end of the storage period. The lower ethylene production rate of melatonin-treated asparagus may be the result of downregulation of the expression of ethylene biosynthesis-related genes [38]. This finding is consistent with reports that melatonin treatment inhibited ethylene production in bananas [39]. Excessive ethylene synthesis leads to the deterioration of vegetables, thereby increasing hydrolytic enzyme activity, which reduces the nutrient and chlorophyll content [5,40,41].

### 3.2. Effect of Melatonin Treatment on Texture

Texture is one of the criteria used for quality assurance and the assessment of asparagus quality deterioration. The firmness differs depending on the site of analysis; the apical part is less firm than the basal part [12]. Strength and toughness are related to fibrousness and the hardening process that occurs after harvesting; the latter is accompanied by lignification of pericyclic (sclerenchyma) fibers. Furthermore, these characteristics may also be associated with respiratory water loss and increases in phenolic compounds, besides lignin [3]. The firmness values for both the melatonin-treated and control samples, measured at the apical and middle parts of asparagus, were in the range of 29.08–34.61 N (Table 1) and did not show significant differences among the sampling days. However, the firmness in the basal part of the control samples increased over the storage period, from 35.16 ±4.13 to 52.20 ± 4.22 N. The basal firmness of 0.5 MLT was significantly lower than that of the control at day 10 of storage; however, the increase in firmness was significantly delayed from day 5 in 1.0 MLT and 2.0 MLT, with day 25 measurements of 40.19 ± 7.55 and 40.71 ± 4.37 N, respectively, i.e., 23.00 and 22.01% lower than the control value, respectively.

### 3.3. Effect of Melatonin Treatment on Color and Chlorophyll Content

Color has a critical effect on consumer acceptability. Therefore, objective color measurements (*L**, *a**, and *b**) were taken to monitor the quality of green asparagus during the storage period (Table 2). The initial *L** values of all the treatment groups were approximately 57. The *L** value of the control decreased gradually until the end of storage, with a value of 48.05 ± 2.65 at day 25. Melatonin treatment significantly delayed the decrease in *L**; 0.5 MLT had relatively high *L** values (>57) for up to day 10, after which it decreased to 51.95 ± 2.64 by day 25, whereas 1.0 MLT and 2.0 MLT maintained *L** values >57 for 15 days, followed by gradual decreases to 53.60 ± 1.67 and 53.23 ± 2.36, respectively, by day 25. At the end of the storage period, the *L** value of 0.5 MLT, 1.0 MLT, and 2.0 MLT was 8.11, 11.55, and 10.78% higher than that of the control group, respectively. These results indicate that melatonin treatment can delay the decrease in the lightness color attribute of asparagus spears during cold storage.

The *a** value relates to the green-red opponent color axis, with negative values ascribed to green and positive values ascribed to red. Therefore, a loss of the green color of asparagus is associated with an increase in *a**. The initial *a** value of all samples ranged between −6.98 and −6.93. There was no significant difference between the *a** values of the control and melatonin treatment groups at day 5 of storage. However, the *a** value of all melatonin treatment groups was significantly lower than that of the control from day 10 until the end of storage, indicating that melatonin treatment delays the loss of green color.

The yellowness (*b** value) of all treatment groups increased with time. There was no significant difference between the *b** value of the control and melatonin treatment during storage for the first 10 days. However, the *b** value of 1.0 MLT was significantly lower than that of the control at day 15; 0.5 MLT and 2.0 MLT had a significantly lower *b** value than the control at day 20. The change in *a** and *b** values of green asparagus was attributed to the degradation of chlorophyll, leading to a decrease in greenness intensity and an increase in yellowness intensity [13].

The degree of color change (Δ*E*) increased with storage time in all treatment groups (Figure 2a). However, the color change in melatonin-treated asparagus at all concentrations was significantly lower than that of the control from day 5. After 25 days of storage, the color change in 0.5 MLT, 1.0 MLT, and 2.0 MLT groups relative to that in the control amounted to 41.39, 52.52, and 50.96%, respectively. The appearance of green asparagus after 25 days of storage is shown in Figure 2b. All the samples showed evidence of wrinkling, mostly in the middle and basal parts. However, the control asparagus showed more wrinkling than melatonin-treated asparagus. Melatonin-treated asparagus (1.0 MLT and 2.0 MLT) had a brighter color than the control asparagus due to less water loss than control, consistent with the total color difference results.

Normally, the chlorophyll content of fresh green asparagus varies widely between 9 and 52 mg/kg FW [5,13,42,43]. The initial chlorophyll concentration depends on several factors, including genetics, environmental conditions, and production methods. However, after harvesting, the chlorophyll content in green asparagus decreases gradually with storage time. Previous studies have shown that the chlorophyll content of green asparagus decreases by 60–66% when stored at 4 °C for 30 days [5]. The initial chlorophyll content of asparagus in this study was approximately 25 mg/Kg FW (Table 3). It declined gradually during the storage period for both the control and melatonin-treated asparagus. There were no differences between the chlorophyll contents of the control and melatonin-treated asparagus on day 0. However, the chlorophyll content of melatonin-treated asparagus was significantly higher than that of the control from day 5 until the end of storage. By day 25, 0.5 MLT, 1.0 MLT, and 2.0 MLT showed a 20.93, 40.41 and 33.57% delay, respectively, in the decrease in chlorophyll content compared with the control. This may be because melatonin functions as an anti-senescence agent, downregulating key chlorophyll catabolism enzymes and other senescence-promoting genes [20]. The decrease in chlorophyll content in green asparagus was consistent with the Δ*E* results. Moreover, this finding is in agreement with that of Wu et al. [22], Miao et al. [44], and Tan et al. [20], who reported a delay in chlorophyll degradation in broccoli florets and Chinese flowering cabbage leaves after melatonin treatment.

### 3.4. Effect of Melatonin Treatment on Vitamin C Content

Vitamin C content can serve as a freshness index of green asparagus because it is known to decrease rapidly with prolonged storage time [45]. The vitamin C content of asparagus was initially in the range of 29.87–30.43 mg/100 g FW (Table 4), decreasing gradually with increasing storage time in both controls and melatonin-treated asparagus. There was no significant difference between vitamin C levels in control and melatonin-treated asparagus up to day 10. However, 100 and 200 μM melatonin-treated asparagus had significantly higher vitamin C content than the control at day 15 of storage. At the end of the storage period, the vitamin C content of 0.5 MLT, 1.0 MLT, and 2.0 MLT was 31.21, 36.25, and 37.73% higher that of the control group, respectively. This result is in agreement with that of Miao et al. [44] and Shekari et al. [46], who reported that melatonin treatment delayed the decrease in vitamin C content during storage of broccoli florets and mushrooms, respectively.

### 3.5. Effect of Melatonin Treatment on Lignin Content

Lignin is a complex polymer of phenylpropanoid residues that imparts rigidity and is mainly deposited in the cell walls [4]. In asparagus shoots, lignin is found primarily in the middle and basal parts; the amount is directly related to freshness. Increased lignin content reduces the desirable sensory characteristics and the market value of asparagus [47]. The initial lignin content of all treatment groups was approximately 30–31 g/Kg FW (Table 5), which increased gradually with time. From day 0 to day 25, the lignin content of controls increased by 139.13% whereas that of 0.5 MLT, 1.0 MLT, and 2.0 MLT increased by 120.00, 90.58, and 93.09%, respectively. The lignification results were consistent with the increase in the firmness and rigidity of the basal part (Table 1) and indicate that melatonin treatment significantly delays lignification of green asparagus during cold storage.

### 3.6. Effect of Melatonin Treatment on H_2_O_2_ Content

H_2_O_2_ is a reactive oxygen species (ROS) with the potential to induce cell membrane damage, which leads to the senescence of fruits and vegetables [38]. The initial H_2_O_2_ content of the controls was 9.03 ± 1.65 mg/100 g FW, whereas that of melatonin-treated asparagus was approximately 8.13–8.73 mg/100 g FW (Table 6). The H_2_O_2_ concentration increased gradually in all the treatment groups during storage. However, melatonin-treated asparagus had significantly lower H_2_O_2_ content than the controls from day 5 to day 25. Higher melatonin concentration was associated with lower H_2_O_2_ content at day 15 of storage. Nevertheless, there was no significant difference between the H_2_O_2_ content of 1.0 MLT and 2.0 MLT at any time point during storage, thus indicating that melatonin treatment reduces the rate of generation of H_2_O_2_. According to Lwin et al. [6], H_2_O_2_ is involved in chlorophyll bleaching when catalyzed by POD at the proper concentration of phenols. Therefore, a lower H_2_O_2_ content reduces chlorophyll degradation and delays quality deterioration. This finding is supported by the color change and chlorophyll degradation assay results (Table 2 and Table 3, respectively).

### 3.7. Effect of Melatonin Treatment on Enzymatic Activity

Lignification is a fundamental developmental process in higher plants and is controlled by several enzymes including PAL, cinnamate 4-hydroxylase, and 4-coumarate-CoA ligase [48]. PAL is the key enzyme involved in the phenylpropanoid and lignin synthesis pathways. A decrease in the overall quality and shelf life of asparagus has been correlated with increased PAL activity [31]. Therefore, understanding PAL activity may help predict the shelf life of asparagus after melatonin treatment. PAL activity (Figure 3a) of the control and 0.5 MLT increased and reached the maximum levels of 99.34 ± 3.12 and 88.78 ± 3.30 U/mg protein, respectively, by day 15, and then gradually decreased to 62.47 ± 2.48 and 62.66 ± 3.88 U/mg protein, respectively, at day 25. Conversely, the PAL activity of 1.0 MLT and 2.0 MLT was significantly lower than that of the control at day 15 by 32.55 and 31.54%, respectively; by day 25, it was 14.53 and 11.76% lower than the control PAL activity, respectively. This finding is consistent with that of the study by Li et al. [7], who reported that melatonin treatment delayed post-harvest lignification of bamboo shoots. Moreover, many studies have reported that inhibition of PAL enzyme activity by post-harvest treatments, such as l-arginine [4], ozone [31], and ammonium sulfate [49] delayed lignin accumulation in asparagus.

POD activity correlates with the degree of lignification in asparagus, as this enzyme promotes the interlinking of lignin precursors following H_2_O_2_ decomposition [29]. POD activity in the asparagus was initially in the range of 28.56–28.99 U/mg protein (Figure 3b). The enzyme activity increased steadily in both controls and melatonin-treated asparagus. However, POD activity was lower in all melatonin-treated asparagus groups than in the control at day 10 of storage. At day 25, the POD activity of 0.5 MLT, 1.0 MLT, and 2.0 MLT was lower than that of the control by 10.41, 17.61, and 17.17%, respectively. These results indicate that melatonin treatment delays lignification in asparagus through the regulation of PAL and POD activity.

There were similar changes in CAT activity of both control and melatonin-treated asparagus (Figure 3c)—i.e., an increase over the first 10 days, followed by a decrease with prolonged storage time. However, CAT activity of all melatonin treatment groups was higher than that of the control at every sampling day; the highest CAT activity (85.33 ± 5.51 U/mg protein) was observed in 1.0 MLT at day 10 of storage. CAT activity of the control decreased sharply from day 10 to day 15 by 39.4%, whereas that of all melatonin treatment groups was nearly the same at day 15 as that at day 10. This observation indicates that melatonin treatment activates CAT at the beginning stage and delays the loss of CAT activity at the later stages.

The change in SOD activity of control and melatonin-treated asparagus (Figure 3d) showed the same trend as the change in CAT activity (Figure 3c). Initially, there was no significant difference between the control and melatonin-treated groups. Thereafter, the SOD activity of the melatonin treatment groups was consistently higher than that of the control group. The highest SOD activity in the control group (43.33 ± 3.51 U/mg protein) was observed at day 10, whereas the highest SOD activity in 0.5 MLT, 1.0 MLT, and 2.0 MLT was observed at day 15, 10, and 10, respectively (i.e., 57.33 ± 3.51, 59.33 ± 1.53, and 58.67 ± 3.06 U/mg protein, respectively). At day 25, SOD activity in 0.5 MLT, 1.0 MLT, and 2.0 MLT were higher than that of the control group by 49.10, 131.57, and 82.47%, respectively. This result indicates that melatonin treatment promotes SOD activity.

ROS promote senescence of fruits and vegetables because they destroy biological macromolecules and affect metabolism, resulting in damages such as increasing cell membrane leakage [38,50]. However, ROS such as H_2_O_2_ and singlet oxygen can be eliminated by CAT and SOD, which protect against free radical attack [29,51]. Therefore, increasing CAT and SOD activity delays post-harvest deterioration of fruits and vegetables. According to the data obtained from the H_2_O_2_ content, CAT activity, and POD activity assays, we suggest that melatonin treatment activates CAT and POD, thereby reducing H_2_O_2_ content and delaying green asparagus deterioration. This result is consistent with that of studies that noted increased CAT and POD activity after melatonin treatment of bamboo shoots, pomegranates, and tomatoes [7,52,53].

## 4. Conclusions

The findings of the present study revealed that melatonin treatment effectively retarded the deterioration of green asparagus quality attributes during a cold-storage period of 25 days, as indicated by a decrease in the percentage weight loss and Ri, as well as a delay in chlorophyll degradation. Melatonin treatment also reduced lignin content, which led to a delay in the increase in firmness via the regulation of PAL and POD activity. Moreover, melatonin treatment increased CAT and POD enzyme activity, leading to reduced H_2_O_2_ content and extended shelf life. We recommend treatment with 100 μM melatonin solution as a practical method to prolong the shelf life of green asparagus. However, melatonin is prone to degradation; therefore, there is a need to develop specialized materials such as nanoporous materials for protecting melatonin from light and air.

## Figures and Tables

**Figure 1 foods-10-02111-f001:**
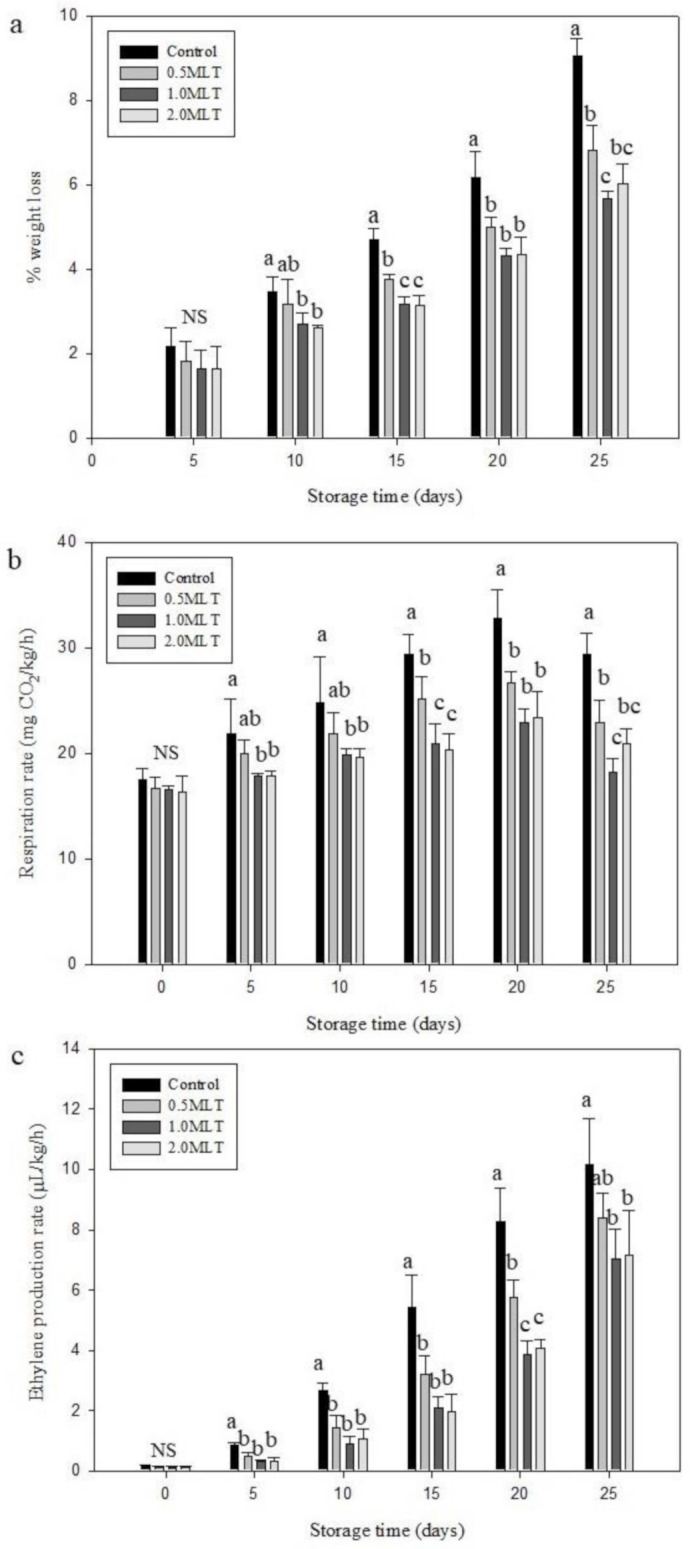
Effect of melatonin treatment on the percentage weight loss (**a**), respiration rate (**b**), and ethylene production rate (**c**) of green asparagus during 25 days of storage at 4 °C and 90% relative humidity. Treatment in melatonin solution: 0.5 MLT (50 μM), 1.0 MLT (100 μM), 2.0 MLT (200 μM); control (distilled water). Values are presented as mean ± standard deviation (*n* = 3). Different letters indicate significant differences (*p* < 0.05); NS = not significant.

**Figure 2 foods-10-02111-f002:**
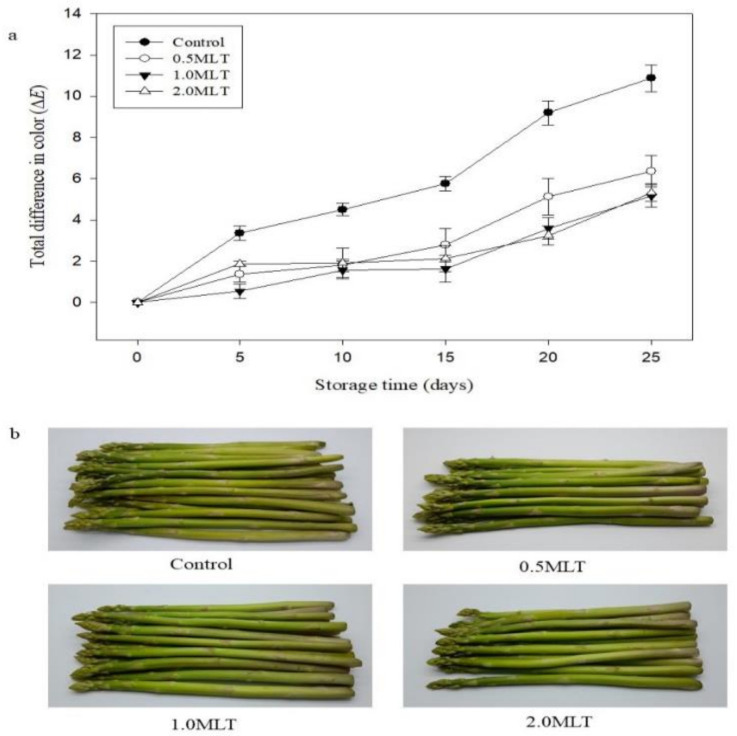
(**a**) Color change (Δ*E*) in green asparagus during 25 days of storage at 4 °C and 90% relative humidity. Treatment in melatonin solution: 0.5 MLT (50 μM), 1.0 MLT (100 μM), 2.0 MLT (200 μM); control (distilled water). (**b**) Appearance of green asparagus after 25 days of storage.

**Figure 3 foods-10-02111-f003:**
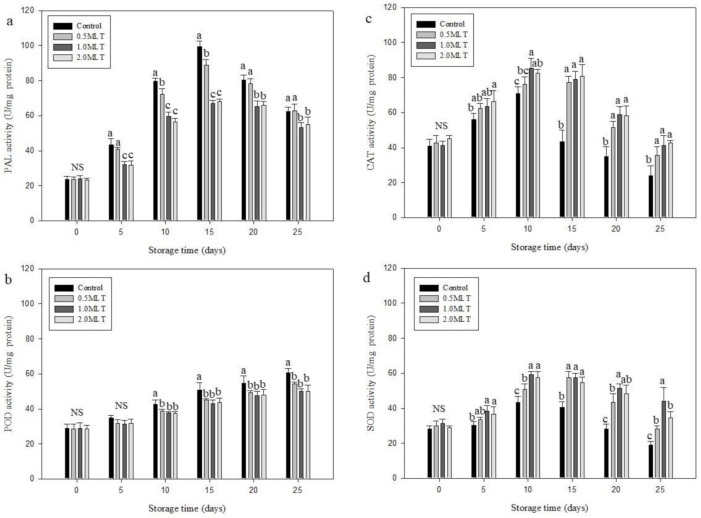
Effect of melatonin treatment on (**a**) phenylalanine ammonia-lyase (PAL) activity, (**b**) peroxidase (POD) activity, (**c**) catalase (CAT) activity, and (**d**) superoxide dismutase (SOD) activity of green asparagus during 25 days of storage at 4 °C and 90% relative humidity. Treatment in melatonin solution: 0.5 MLT (50 μM), 1.0 MLT (100 μM), 2.0 MLT (200 μM); control (distilled water). Values are presented as mean ± standard deviation (*n* = 3). Different letters indicate significant differences (*p* < 0.05); NS = not significant.

**Table 1 foods-10-02111-t001:** Effect of melatonin treatment on firmness (N) at the apical, middle, and basal parts of green asparagus during 25 days of storage at 4 °C and 90% relative humidity.

Part of Asparagus	Treatment	Storage Time (Days)					
		0	5	10	15	20	25
Apical	Control	31.21 ± 3.03 ^a,A^	30.80 ± 3.84 ^a,A^	33.07 ± 3.07 ^a,A^	30.24 ± 2.47 ^a,A^	34.61 ± 6.39 ^a,A^	33.51 ± 7.28 ^a,A^
	0.5 MLT	31.56 ± 2.43 ^a,A^	32.28 ± 2.31 ^a,A^	30.17 ± 4.48 ^a,A^	31.73 ± 3.39 ^a,A^	33.27 ± 5.02 ^a,A^	33.43 ± 4.31 ^a,A^
	1.0 MLT	31.39 ± 2.95 ^a,A^	30.22 ± 4.12 ^a,A^	32.52 ± 3.49 ^a,A^	30.06 ± 4.21 ^a,A^	34.42 ± 6.23 ^a,A^	33.92 ± 3.65 ^a,A^
	2.0 MLT	30.32 ± 3.13 ^a,A^	33.41 ± 2.00 ^a,A^	32.09 ± 4.17 ^a,A^	31.49 ± 3.95 ^a,A^	33.95 ± 4.50 ^a,A^	34.52 ± 3.84 ^a,A^
Middle	Control	31.90 ± 3.13 ^a,A^	29.32 ± 3.67 ^a,A^	34.17 ± 3.37 ^a,A^	33.42 ± 3.16 ^a,A^	33.52 ± 4.67 ^a,A^	29.55 ± 2.30 ^a,A^
	0.5 MLT	32.40 ± 2.71 ^a,A^	29.08 ± 4.59 ^a,A^	30.08 ± 4.65 ^a,A^	30.56 ± 4.11 ^a,A^	31.41 ± 6.13 ^a,A^	29.81 ± 2.05 ^a,A^
	1.0 MLT	32.50 ± 3.81 ^a,A^	30.02 ± 4.44 ^a,A^	30.33 ± 4.70 ^a,A^	31.44 ± 4.58 ^a,A^	31.30 ± 3.96 ^a,A^	30.31 ± 4.71 ^a,A^
	2.0 MLT	31.20 ± 3.06 ^a,A^	31.55 ± 6.83 ^a,A^	31.38 ± 3.21 ^a,A^	32.55 ± 6.29 ^a,A^	32.41 ± 4.48 ^a,A^	29.65 ± 4.29 ^a,A^
Basal	Control	35.46 ± 4.13 ^a,^^A^	41.36 ± 5.35 ^c,B^	46.82 ± 5.96 ^c,C^	48.66 ± 6.40 ^b,D^	50.06 ± 2.16 ^c,E^	52.20 ± 4.22 ^c,F^
	0.5 MLT	36.37 ± 2.03 ^a,A^	38.35 ± 4.14 ^bc,A^	40.54 ± 4.34 ^b,B^	43.76 ± 8.89 ^ab,C^	45.38 ± 4.05 ^b,D^	46.50 ± 5.28 ^b,D^
	1.0 MLT	35.49 ± 3.45 ^a,A^	34.88 ± 5.33 ^ab,A^	35.74 ± 6.09 ^a,A^	37.73 ± 3.72 ^a,B^	39.37 ± 3.66 ^a,C^	40.19 ± 7.55 ^a,C^
	2.0 MLT	35.16 ± 4.81 ^a,A^	34.16 ± 4.96 ^a,A^	35.36 ± 4.19 ^a,A^	38.40 ± 5.15 ^a,B^	39.56 ± 4.48 ^a,C^	40.71 ± 4.37 ^a,C^

Treatment in melatonin solution: 0.5 MLT (50 μM), 1.0 MLT (100 μM), 2.0 MLT (200 μM); control (distilled water). Firmness values (N) are presented as mean ± standard deviation (*n* = 10). Different superscript lowercase letters in the same column of each part of asparagus indicate significant differences (*p* < 0.05); Different superscript uppercase letters in the same row indicate significant differences (*p* < 0.05).

**Table 2 foods-10-02111-t002:** Effect of melatonin treatment on color attributes (*L**, *a**, *b**) of green asparagus during 25 days of storage at 4 °C and 90% relative humidity.

Color Parameter	Treatment	Storage Time (Days)					
		0	5	10	15	20	25
*L**	Control	57.30 ± 1.60 ^a,A^	54.56 ± 3.32 ^a,B^	53.42 ± 1.52 ^a,B^	52.94 ± 2.09 ^a,B^	49.28 ± 2.20 ^a,C^	48.05 ± 2.65 ^a,C^
	0.5 MLT	57.35 ± 1.87 ^a,AB^	58.89 ± 2.15 ^b,A^	57.31 ± 1.67 ^b,AB^	55.59 ± 1.10 ^a,B^	52.99 ± 1.34 ^b,C^	51.95 ± 2.64 ^b,C^
	1.0 MLT	57.82 ± 1.60 ^a,A^	58.19 ± 3.38 ^b,A^	58.87 ± 2.25 ^b,A^	57.46 ± 2.18 ^c,A^	54.81 ± 2.62 ^bc,B^	53.60 ± 1.67 ^b,B^
	2.0 MLT	57.52 ± 1.24 ^a,AB^	58.99 ± 3.31 ^b,A^	58.78 ± 2.36 ^b,A^	57.02 ± 1.96 ^bc,AB^	55.27 ± 2.93 ^c,BC^	53.23 ± 2.36 ^b,C^
*a**	Control	−6.93 ± 0.59 ^a,A^	−6.11 ± 1.36 ^a,B^	−5.77 ± 0.93 ^a,BC^	−5.68 ± 0.42 ^a,BC^	−5.35 ± 0.70 ^a,BC^	−5.31 ± 0.94 ^a,C^
	0.5 MLT	−6.99 ± 0.57 ^a,A^	−6.58 ± 0.83 ^a,AB^	−6.57 ± 0.58 ^b,AB^	−6.20 ± 0.81 ^b,BC^	−5.84 ± 1.12 ^ab,BC^	−5.58 ± 0.80 ^b,C^
	1.0 MLT	−6.98 ± 0.52 ^a,A^	−6.75 ± 0.40 ^a,AB^	−6.72 ± 0.94 ^b,AB^	−6.49 ± 0.80 ^b,ABC^	−6.18 ± 0.52 ^b,BC^	−5.99 ± 0.30 ^b,C^
	2.0 MLT	−6.93 ± 0.40 ^a,A^	−6.93 ± 0.40 ^a,A^	−6.60 ± 1.06 ^b,AB^	−6.31 ± 0.31 ^b,BC^	−5.95 ± 0.29 ^ab,CD^	−5.54 ± 0.47 ^b,D^
*b**	Control	32.09 ± 1.45 ^a,A^	33.98 ± 3.06 ^a,AB^	34.17 ± 1.42 ^a,AB^	35.74 ± 2.90 ^b,BC^	36.45 ± 2.42 ^b,C^	37.67 ± 2.31 ^b,C^
	0.5 MLT	32.19 ± 1.38 ^a,A^	32.67 ± 2.95 ^a,AB^	33.94 ± 2.18 ^a,AB^	34.21 ± 1.47 ^ab,BC^	34.63 ± 1.11 ^a,C^	35.26 ± 2.24 ^a,C^
	1.0 MLT	31.62 ± 1.24 ^a,A^	32.08 ± 1.59 ^a,A^	33.38 ± 1.85 ^a,AB^	33.65 ± 2.03 ^a,AB^	33.84 ± 1.77 ^a,B^	34.84 ± 2.12 ^a,B^
	2.0 MLT	32.14 ± 1.44 ^a,A^	33.27 ± 2.12 ^a,AB^	33.55 ± 2.54 ^a,AB^	34.12 ± 1.94 ^ab,AB^	34.27 ± 1.99 ^a,AB^	34.98 ± 3.33 ^a,B^

Treatment in melatonin solution: 0.5 MLT (50 μM), 1.0 MLT (100 μM), 2.0 MLT (200 μM); control (distilled water). Values are presented as mean ± standard deviation (*n* = 10). Different superscript lowercase letters in the same column indicate significant differences (*p* < 0.05); Different superscript uppercase letters in the same row indicate significant differences (*p* < 0.05).

**Table 3 foods-10-02111-t003:** Effect of melatonin treatment on chlorophyll content (mg/Kg fresh weight) of green asparagus during 25 days of storage at 4 °C and 90% relative humidity.

Treatment	Storage Time (Days)					
	0	5	10	15	20	25
Control	25.75 ± 0.63 ^a,A^	20.81 ± 0.94 ^a,B^	19.77 ± 1.05 ^a,BC^	18.30 ± 0.96 ^a,CD^	17.50 ± 0.49 ^a,D^	15.07 ± 0.39 ^a,E^
0.5 MLT	25.13 ± 0.46 ^a,A^	23.55 ± 1.26 ^b,B^	22.53 ± 0.92 ^b,BC^	21.77 ± 0.51 ^b,CD^	20.53 ± 0.68 ^b,D^	19.06 ± 0.53 ^b,E^
1.0 MLT	25.90 ± 0.49 ^a,A^	24.38 ± 0.26 ^b,B^	23.20 ± 0.40 ^b,C^	22.79 ± 0.76 ^b,C^	22.12 ± 0.67 ^c,CD^	21.16 ± 0.73 ^d,E^
2.0 MLT	25.41 ± 0.92 ^a,A^	24.20 ± 0.43 ^b,B^	23.10 ± 0.46 ^b,BC^	22.24 ± 0.54 ^b,C^	20.48 ± 0.96 ^b,D^	20.13 ± 0.44 ^c,D^

Treatment in melatonin solution: 0.5 MLT (50 μM), 1.0 MLT (100 μM), 2.0 MLT (200 μM); control (distilled water). Chlorophyll content values (mg/kg fresh weight) are presented as mean ± standard deviation (*n* = 3). Different superscript lowercase letters in the same column indicate significant differences (*p* < 0.05); Different superscript uppercase letters in the same row indicate significant differences (*p* < 0.05).

**Table 4 foods-10-02111-t004:** Effect of melatonin treatment on vitamin C content (mg/100 g fresh weight) of green asparagus during 25 days of storage at 4 °C and 90% relative humidity.

Treatment	Storage Time (Days)					
	0	5	10	15	20	25
Control	30.20 ± 1.18 ^a,A^	29.23 ± 1.75 ^a,A^	25.26 ± 2.12 ^a,B^	18.38 ± 1.10 ^a,C^	16.04 ± 1.10 ^a,C^	12.88 ± 1.53 ^a,D^
0.5 MLT	29.87 ± 1.19 ^a,A^	29.30 ± 2.86 ^a,AB^	26.64 ± 0.63 ^a,B^	21.22 ± 2.02 ^ab,C^	18.49 ± 1.56 ^b,CD^	16.90 ± 0.80 ^b,D^
1.0 MLT	30.43 ± 1.33 ^a,A^	31.33 ± 2.60 ^a,A^	26.78 ± 0.70 ^a,B^	22.33 ± 2.02 ^b,C^	20.54 ± 0.80 ^b,C^	17.55 ± 1.32 ^b,D^
2.0 MLT	29.92 ± 0.52 ^a,A^	30.57 ± 1.97 ^a,A^	26.50 ± 1.19 ^a,B^	22.58 ± 0.95 ^b,C^	20.57 ± 0.66 ^b,C^	17.74 ± 2.20 ^b,D^

Treatment in melatonin solution: 0.5 MLT (50 μM), 1.0 MLT (100 μM), 2.0 MLT (200 μM); control (distilled water). Vitamin C content values (mg/100 g fresh weight) are presented as mean ± standard deviation (*n* = 3). Different superscript lowercase letters in the same column indicate significant differences (*p* < 0.05); Different superscript uppercase letters in the same row indicate significant differences (*p* < 0.05).

**Table 5 foods-10-02111-t005:** Effect of melatonin treatment on lignin content (g/Kg fresh weight) of green asparagus during 25 days of storage at 4 °C and 90% relative humidity.

Treatment	Storage Time (Days)					
	0	5	10	15	20	25
Control	31.33 ± 3.21 ^a,A^	40.91 ± 3.31 ^b,B^	50.85 ± 1.44 ^c,C^	58.58 ± 1.08 ^c,D^	69.35 ± 1.08 ^c,E^	74.92 ± 2.97 ^c,F^
0.5 MLT	30.71 ± 2.16 ^a,A^	36.17 ± 3.15 ^ab,B^	42.97 ± 3.06 ^b,C^	51.94 ± 2.11 ^b,D^	59.20 ± 1.44 ^b,E^	67.57 ± 2.97 ^b,F^
1.0 MLT	31.23 ± 3.88 ^a,A^	34.07 ± 1.26 ^a,A^	38.69 ± 0.74 ^a,B^	42.51 ± 1.15 ^a,BC^	45.46 ± 2.54 ^a,C^	59.52 ± 1.68 ^a,D^
2.0 MLT	30.86 ± 4.90 ^a,A^	34.20 ± 2.05 ^a,AB^	38.33 ± 1.65 ^a,BC^	42.33 ± 1.23 ^a,CD^	45.54 ± 3.45 ^a,D^	59.59 ± 3.47 ^a,E^

Treatment in melatonin solution: 0.5 MLT (50 μM), 1.0 MLT (100 μM), 2.0 MLT (200 μM); control (distilled water). Lignin content values (g/100 mg fresh weight) are presented as mean ± standard deviation (*n* = 3). Different superscript lowercase letters in the same column indicate significant differences (*p* < 0.05); Different superscript uppercase letters in the same row indicate significant differences (*p* < 0.05).

**Table 6 foods-10-02111-t006:** Effect of melatonin treatment on hydrogen peroxide content (mg/100 g fresh weight) of green asparagus during 25 days of storage at 4 °C and 90% relative humidity.

Treatment	Storage Time (Days)					
	0	5	10	15	20	25
Control	9.03 ± 1.65 ^a,A^	13.33 ± 0.67 ^b,B^	15.33 ± 1.00 ^c,C^	16.63 ± 1.30 ^c,C^	19.33 ± 0.68 ^c,D^	22.43 ± 0.59 ^c,D^
0.5 MLT	8.13 ± 0.67 ^a,A^	9.90 ± 0.19 ^a,AB^	11.60 ± 0.67 ^b,BC^	12.70 ± 1.67 ^b,C^	15.28 ± 0.95 ^b,D^	17.33 ± 0.80 ^b,D^
1.0 MLT	8.23 ± 1.06 ^a,A^	8.93 ± 0.81 ^a,AB^	9.57 ± 0.7 ^ab,ABC^	10.07 ± 0.99 ^a,BC^	11.07 ± 0.59 ^a,C^	14.20 ± 0.70 ^a,C^
2.0 MLT	8.73 ± 0.67 ^a,A^	8.48 ± 1.20 ^a,A^	9.00 ± 0.70 ^a,A^	10.47 ± 0.31 ^a,AB^	12.17 ± 1.88 ^a,B^	14.57 ± 0.55 ^a,B^

Treatment in melatonin solution: 0.5 MLT (50 μM), 1.0 MLT (100 μM), 2.0 MLT (200 μM); control (distilled water). Hydrogen peroxide content values (mg/100 mg fresh weight) values are presented as mean ± standard deviation (*n* = 3). Different superscript lowercase letters in the same column indicate significant differences (*p* < 0.05); Different superscript uppercase letters in the same row indicate significant differences (*p* < 0.05).

## Data Availability

The data supporting the findings of this study are included in this article.

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
