# Peer review of "Exogenous Melatonin Reduces Lignification and Retains Quality of Green Asparagus (Asparagus officinalis L.)"

_foods, 2021, doi:10.3390/foods10092111_

Round 1
Reviewer 1 Report
see the attached file

Author Response
Please check my answer file about the reviewer's comments.
Thanks

Reviewer 2 Report
In the graphical abstract, I suggest to use one decimal for %. Moreover, please write Reduction (of…) and Delay of firmness increase) and Enhancement (of…)
Please change “results compared between” into “comparison between”
Lines 40-41: please rephrase: “texture is an important freshness indicator of asparagus spears”
Line 42: “after harvesting, the firmness of asparagus increases” (being a general behaviour of asparagus).
Line 48: please add another very recent reference among the packaging strategies: https://doi.org/10.3390/foods10020478. This reference could offer suitable data comparison for the tested parameters.
Line 85: are coded as: 0.1 MLT …
Par. 2.3.2. Please specify the type of GC column used.
Lines 109-110. Please specify how ethylene peak areas were converted to concentration (external calibration?)
Line 117. Please check this detail: 1 mm diameter shear blade. Is it diameter of thickness?
In par. 3.2 authors say that differences in texture during storage were not significant in apical and middle parts, while they were significant for the basal part. Statistical analysis would be useful to assess the significance of differences among storage times. I suggest that the authors remove “NS” (if letters are not reported, it means that differences are not significant), and use capital letters for comparison among treatments (in column) and small letters for comparison among storage times (in line).
Tables 2-6. As for table 1, I suggest that the authors analyze differences among storage times. Please, remove “NS” (if letters are not reported, it means that differences are not significant), and use capital letters for comparison among treatments (in column) and small letters for comparison among storage times (in line).
Lines 344-345. The first sentence could be eliminated, has no relevance based on the paper topic.
Authors could attempt to calculate correlation between firmness (N) and lignin data.
Author Response
Please check my answer file for the reviewer's comments.
Thanks,
